# A Result Regarding Finite-Time Stability for Hilfer Fractional Stochastic Differential Equations with Delay

**Man Li** [1], **Yujun Niu** [1] **and Jing Zou** [2],*

---

[1] School of Mathematics and Physics, Nanyang Institute of Technology, Nanyang 473004, China; manabcli@163.com (M.L.); nyjyrff@yeah.net (Y.N.)

[2] Department of Mathematics, Guizhou University, Guiyang 550025, China

* Correspondence: zjing210107@163.com

**Abstract:** Hilfer fractional stochastic differential equations with delay are discussed in this paper. Firstly, the solutions to the corresponding equations are given using the Laplace transformation and its inverse. Afterwards, the Picard iteration technique and the contradiction method are brought up to demonstrate the existence and uniqueness of understanding, respectively. Further, finite-time stability is obtained using the generalized Grönwall–Bellman inequality. As verification, an example is provided to support the theoretical results.

**Keywords:** stochastic differential equations; fractional calculus; delay; existence and uniqueness; finite-time stability





## 1. Introduction

Fractional calculus is favored by many researchers because of its genetic characteristics. In simulation processes related to control theory, physics, chaos and turbulence, fluid mechanics and visco-elastic materials, fractional differential equations obtain better results than integer-order differential equations. For related content, readers can refer to the monographs [1–4]. Therefore, in recent decades, fractional calculus has gradually become a powerful tool for discussing and resolving the problems of modern production technology in relation to the rapid development of such technology and natural science. It is worth noting that, in the study of control theory, some researchers use fractional calculus to obtain better simulation results [5]. There are also some related findings that can be referred to in the articles [6–8]. Significantly, as an improvement of Riemann–Liouville fractional calculus and Caputo fractional calculus, Hilfer fractional calculus is more widely used in practical life. Gou [9] studied the monotonic iterative technique for a kind of Hilfer fractional system.

Random disturbances are well-known and unavoidable factors in the study of real systems. They are also one of the important factors in research on system stability and play an indispensable role. Fractional Brownian motion is widely used to describe uncertainty because of its excellent properties, such as self-similarity and long-distance correlation. In many fields of stochastic analysis, fractional Brownian motion has attracted much attention because of these good properties. On this basis, investigators have carried out many interesting studies and obtained many useful conclusions [10–21]. In general, the study of stochastic systems is very meaningful and challenging.

In order to get closer to real life, investigators tend to study systems with delays in their research processes. A system is affected by its current state and its past state as well. In general, delay often causes a system to oscillate and become unstable. Therefore, it is necessary to study systems with delay. Some researchers have studied systems with delays and achieved some results. For example, Luo et al. [16] analyzed a class of stochastic fractional differential equations with time delays and proved the results obtained using numerical simulation. Xu et al. [22] studied a class of stochastic delay fractional differential equations powered by Brownian motion. Ahmed et al. investigated differential equations

with Poisson jumps, which are a kind of delay stochastic system with Hilfer-type fractional derivatives, in [23]. Salem et al. [24] established integral representations of solutions for homogeneous and inhomogeneous delay systems for Hilfer fractional derivatives and gave the results in terms of finite-time stability for the corresponding solutions.

So far, there is much more research on deterministic fractional systems with delay than on stochastic fractional systems with delay, and there are many issues to be further studied. In the exploration of fractional stochastic systems, most researchers pay more attention to the existence and uniqueness of solutions (readers can consult [25–28]). With the advance of time, research on the uniqueness and existence of solutions to systems is getting deeper and deeper. On this basis, Ahmed et al. [29] studied a class of Hilfer-type stochastic fractional integro-differential equations and obtained the existence result using the methods of fractional calculus, the Sadovskii fixed-point theorem, and the semigroup property. Johnson et al. discussed a class of infinite delay neutral stochastic Hilfer-type fractional integro-differential equations and demonstrated the existence result for the solutions through the use of Banach fixed-point theory in [30]. By using Hilfer fractional calculus, the fixed-point method, and the Mittag–Leffer function, Gao and Yu [31] studied the uniqueness and existence of the nonlocal values of solutions for a kind of semi-linear system with Hilfer fractional derivatives. Li et al. [32] first established the equivalent Volterra integral equation, and then existence and uniqueness for a kind of fractional system of the Hilfer type with variable coefficients were discussed.

As we all know, a large part of a simulation in practical applications is affected by the dynamic behavior of the model, especially the stability. Therefore, the study of the stability of differential equations has been favored by researchers, and some research results have been achieved [33]. In fact, as early as the 1960s, there were studies on finite-time stability [34]. It is worth mentioning that Dorato [34] studied a kind of time-varying linear system and discussed its short-time stability. Moreover, Kushner [35] further extended the finite-time stability of stochastic systems on the grounds discussed by Dorato. Luo et al. first derived the solution for a system by means of the Laplace transformation and its inverse and then derived the finite-time stability result by using interval translation techniques and the Henry Grönwall delay inequality in [36]. In [37], Coppel's inequality and the Jensen inequality were used to analyze the finite-time stability results for a kind of system with finite delay. There are more finite-time stability results available in [38–42].

As the above discussion shows, to the best of our knowledge, there are not many studies on finite-time stability for Hilfer-type fractional stochastic differential equations with delay. Therefore, we explore the finite-time stability of such systems in this paper.

$$\begin{cases} D_{0^+}^{\gamma,\delta}\chi(\hbar) = A\chi(\hbar) + \kappa(\hbar, \chi(\hbar), \chi(\hbar - \tau)) + \sigma(\hbar, \chi(\hbar), \chi(\hbar - \tau))\frac{dB_\hbar}{d\hbar}, \hbar \in [0, V], \\ \chi(\hbar) = \Psi(\hbar), -\tau \leq \hbar \leq 0, \\ I_{0^+}^{1-\lambda}\chi(0) = 0, \lambda = \gamma + \delta - \gamma\delta, \end{cases} \quad (1)$$

where $D_{0^+}^{\gamma,\delta}$ expresses the Hilfer fractional derivative equipped with $0 \leq \gamma \leq 1$, $0 < \delta < 1$. $A \in \mathbb{R}^{d \times d}$, $\kappa: [0, V] \times \mathbb{R}^d \times \mathbb{R}^d \to \mathbb{R}^d$, and $\sigma: [0, V] \times \mathbb{R}^d \times \mathbb{R}^d \to \mathbb{R}^{d \times n}$ are all continuous measurable functions. $B_\hbar$ is $n$-dimensional Brownian motion defined on a complete probability space $(\Omega, \mathcal{F}, P)$. $\Psi: [-\tau, 0] \to \mathbb{R}^d$ is a continuous function. Let the norm of $\mathbb{R}^d$ be $\| \cdot \|$ and satisfy $\mathbb{E}\|\Psi(\hbar)\|^2 < \infty$.

The major contributions of this paper can be described as the following:

(i) The system we encountered is almost affected by the current states and the past states. Compared with [12,14], System (1) contains delay, which brings the states of the system closer to the states in real life and is more convenient for practical applications;

(ii) The model mentioned in this paper is more general than the model in [41]. The delay term is considered in a fractional stochastic system of the Hilfer type. There is relatively little research on this type of system;

(iii) The difference between this manuscript and [16–19] is that we adopt the method of the Laplace transformation and its inverse when we derive the solutions to the fractional delay stochastic differential equations, and the Mittag–Leffler function is included in the derived processes, which is helpful for subsequent derivations.

The rest of the paper is arranged as follows. We provide some basic preparatory work in Section 2. In Section 3, we first give the solution to the system under consideration using the Laplace transformation and its inverse and then prove the existence and uniqueness of the solution using the Picard iteration technique and the contradiction method. The results for finite-time stability are given in Section 4. Finally, an example is provided to demonstrate the theoretical results in Section 5.

## 2. Essential Definitions and Lemmas

**Definition 1** ([15]). *For a function $\sigma$, the fractional integral operator of order $\delta$ can be described as*

$$I^\delta \sigma(\hbar) = \frac{1}{\Gamma(\delta)} \int_0^\hbar \frac{\sigma(\varpi)}{(\hbar - \varpi)^{1-\delta}} d\varpi, \quad \hbar > 0,$$

*where $\Gamma(\cdot)$ denotes the Gamma function.*

**Definition 2** ([15]). *For $0 \leq \gamma \leq 1$, $0 < \delta < 1$, and a function $\sigma$, the Hilfer-type fractional derivative of orders $\gamma$ and $\delta$ can be described as*

$$D_{0+}^{\gamma,\delta} \sigma(\hbar) = I_{0+}^{\gamma(1-\delta)} \frac{d}{dt} I_{0+}^{(1-\gamma)(1-\delta)} \sigma(\hbar),$$

*where $D = \frac{d}{dt}$.*

**Definition 3** ([15]). *The form of the Mittag–Leffler function can be defined as*

$$E_{b,m}(\xi) = \sum_{i=0}^\infty \frac{\xi^i}{\Gamma(ib + m)}, b, m > 0.$$

*In particular, we have*

$$E_{b,1}(\xi) = \sum_{i=0}^\infty \frac{\xi^i}{\Gamma(ib + 1)} = E_b(\xi).$$

**Definition 4** ([36]). *Suppose the functions $\varphi(\hbar)$ and $\chi(\hbar)$ both satisfy the condition that when $\hbar \to 0$, $\varphi(\hbar) = \chi(\hbar) = 0$; that means $\int_0^\hbar \varphi(\hbar - \iota)\chi(\hbar)d\iota$ is the convolution of $\varphi(\hbar)$ and $\chi(\hbar)$. This is expressed as*

$$\varphi(\hbar) * \chi(\hbar) = \chi(\hbar) * \varphi(\hbar) = \int_0^\hbar \varphi(\hbar - \iota)\chi(\hbar)d\iota.$$

**Lemma 1** ([43]). *For the function $\chi(\hbar)$, the Laplace transformation for the Hilfer fractional derivative can be expressed as*

$$L\{D_{0+}^{\gamma,\delta} \chi(\hbar); t\} = t^\gamma \chi_L(t) - t^{\delta(\gamma-1)} I_{0+}^{1-\lambda} \chi(0),$$

*where $\lambda = \gamma + \delta - \gamma\delta$.*

**Proposition 1.** *The Laplace transformation has some properties, as shown below*

$$L\{\pi(\hbar) * \delta(\hbar); \varpi\} = \Pi(\varpi)\Delta(\varpi);$$

$$L^{-1}\{\Pi(\varpi) * \Delta(\varpi); \varpi\} = \pi(\hbar)\delta(\hbar);$$

$$L^{-1}\{\Pi(\varpi)\Delta(\varpi);\varpi\} = \pi(\hbar) * \delta(\hbar).$$

*Functions $\pi(\hbar)$ and $\delta(\hbar)$ are real-valued functions defined on $[0,\infty)$, and $\Pi(\varpi) = \int_0^\infty \pi(\hbar)$ $e^{-\varpi\hbar}d\hbar$ and $\Delta(\varpi) = \int_0^\infty \delta(\hbar)e^{-\varpi\hbar}d\hbar$ are called image functions of $\pi(\hbar)$ and $\delta(\hbar)$, respectively.*

**Lemma 2** ([15]). *For the Mittag–Leffler function, the Laplace transformation can be denoted in the following form*

$$L\{\hbar^{bi+m-1}E_{b,m}^{(i)}(\pm\beta\hbar^b);t\} = \int_0^\infty e^{-t\hbar}\hbar^{bi+m-1}E_{b,m}^{(i)}(\pm\beta\hbar^b)d\hbar$$

$$= \frac{i!t^{b-m}}{(t^b \mp \beta)^{i+1}}.$$

**Lemma 3** ([44]). *Given a random variable $\omega$, $\forall l > 0$ and $1 \leq m < \infty$, and we can deduce that the formula mentioned below is true*

$$\mathbb{P}(\|\omega\| \geq l) \leq \frac{1}{l^m}\mathbb{E}(\|\omega\|^m).$$

**Lemma 4** ([44]). *If $E_n$ satisfies $\{E_n\} \subset \mathcal{F}$ and $\sum\limits_{n=1}^\infty \mathbb{P}(E_n) < \infty$, then we can conclude that*

$$\mathbb{P}(\limsup_{n\to\infty} E_n) = 0.$$

**Definition 5** ([36]). *The positive constants $\varrho, \xi,$ and $V$ satisfy $\varrho < \xi$. Thus, the system is finite-time-stable on $[-\tau, V]$ if, when $\mathbb{E}\|\Psi(0)\|^2 \leq \varrho$, we have $\mathbb{E}\|\psi\|^2 \leq \xi$.*

We postulate the following hypotheses to facilitate the smooth development of the following work.

- $(H_1)$ For $\kappa, \sigma$ in System (1), for $\forall\varphi_i, \widehat{\varphi}_i \in \mathbb{R}^d, (i = 1, 2), \hbar \in [-\tau, V]$, and we can find a corresponding constant $\mu_1 > 0$ such that

$$\|\kappa(\hbar, \varphi_1, \widehat{\varphi}_1) - \kappa(\hbar, \varphi_2, \widehat{\varphi}_2)\|^2 \vee \|\sigma(\hbar, \varphi_1, \widehat{\varphi}_1) - \sigma(\hbar, \varphi_2, \widehat{\varphi}_2)\|^2$$
$$\leq \mu_1(\|\varphi_1 - \varphi_2\|^2 + \|\widehat{\varphi}_1 - \widehat{\varphi}_2\|^2).$$

- $(H_2)$ For $\kappa, \sigma$ in System (1), $\varphi, \widehat{\varphi} \in \mathbb{R}^d, \hbar \in [-\tau, V]$, and we can find a constant $\mu_2 > 0$,

$$\|\kappa(\hbar, \varphi, \widehat{\varphi})\|^2 \vee \|\sigma(\hbar, \varphi, \widehat{\varphi})\|^2 \leq \mu_2\Big(1 + \|\varphi\|^2 + \|\widehat{\varphi}\|^2\Big).$$

**Lemma 5** ([16]). *Let $\vartheta_1, \vartheta_2, \cdots, \vartheta_n (n \in N)$ be real numbers and satisfy $\vartheta_i \geq 0, (i = 1, 2, \ldots, n)$. Thus*
$$\left(\sum_{i=1}^n \vartheta_i\right)^p \leq n^{p-1} \sum_{i=1}^n \vartheta_i^p, \quad \text{for } p > 1.$$

**Lemma 6** ([44]). *(Generalized Grönwall–Bellman inequality) Assume $0 < \beta < 1$ and $\hbar \in [0, V)$, where $V \leq \infty$. For $\hbar \in [0, V)$, the locally integrable non-negative functions $x(\hbar), y(\hbar),$ and $z(\hbar)$ are non-negative and nondecreasing bounded continuous functions. $\chi(\hbar)$ is a non-negative and locally integrable function on $[0, V)$ such that*

$$\chi(\hbar) \leq x(\hbar) + y(\hbar)\int_0^\hbar \chi(\varpi)d\varpi + z(\hbar)\int_0^\hbar (\hbar - \varpi)^{\beta-1}\chi(\varpi)d\varpi.$$

*Thus, the following estimation is valid*

$$\chi(\hbar) \leq x(\hbar) + \sum_{n=1}^{\infty} \sum_{i=0}^{n} \binom{n}{i} y^{n-i}(\hbar) \frac{[z(\hbar)\Gamma(\beta)]^i}{\Gamma(i\beta + n - i)} \int_0^{\hbar} (\hbar - \varpi)^{\{i(2\beta - 1) - (i+1-n)\}} x(\varpi) d\varpi.$$

## 3. Existence and Uniqueness

In this section, the equivalent form of system (1) under consideration is derived by means of the Laplace transformation and its inverse and expressed with the Mittag–Leffler function.

**Lemma 7.** *If a function $\chi(\cdot)$ is the solution to the following integral equations, it is also said to be the solution to System* (1)

$$\chi(\hbar) = \begin{cases} \Psi(\hbar), \hbar \in [-\tau, 0], \\ \int_0^{\hbar} (\hbar - \varpi)^{\gamma - 1} E_{\gamma, \gamma}(A(\hbar - \varpi)^{\gamma}) \kappa(\varpi, \chi(\varpi), \chi(\varpi - \tau)) d\varpi \\ + \int_0^{\hbar} (\hbar - \varpi)^{\gamma - 1} E_{\gamma, \gamma}(A(\hbar - \varpi)^{\gamma}) \sigma(\varpi, \chi(\varpi), \chi(\varpi - \tau)) dB_{\varpi}, \\ \varpi \in [0, V]. \end{cases} \qquad (2)$$

**Proof.** For $\hbar \in [-\tau, 0]$, we have

$$\chi(\hbar) = \Psi(\hbar).$$

Taking the Laplace transformation of both sides of System (1) for $\hbar(\hbar \in [0, V])$, we then obtain

$$L\{D_{0+}^{\gamma, \delta} \chi(\hbar); t\} = t^{\gamma} \chi_L(t),$$

followed by

$$t^{\gamma} \chi_L(t) = A\chi_L(t) + \Xi(t) + \Lambda(t),$$

where $\chi_L(t), \Xi(t)$, and $\Lambda(t)$ represent the Laplace transformation of $\chi(\hbar), \kappa(\hbar, \chi(\hbar), \chi(\hbar - \tau))$, and $\sigma(\hbar, \chi(\hbar), \chi(\hbar - \tau)) \frac{dB_{\hbar}}{d\hbar}$, respectively. Therefore

$$\chi_L(t) = \frac{\Xi(t)}{t^{\gamma} - A} + \frac{\Lambda(t)}{t^{\gamma} - A}.$$

If we take the inverse Laplace transformation of both sides of the above formula, then we can obtain the following form

$$\begin{aligned} \chi(\hbar) =& \kappa(\hbar, \chi(\hbar), \chi(\hbar - \tau)) * \hbar^{\gamma - 1} E_{\gamma, \gamma}(A\hbar^{\gamma}) \\ & + \hbar^{\gamma - 1} E_{\gamma, \gamma}(A\hbar^{\gamma}) * \sigma(\hbar, \chi(\hbar), \chi(\hbar - \tau)) \frac{dB_{\hbar}}{d\hbar} \\ =& \int_0^{\hbar} (\hbar - \varpi)^{\gamma - 1} E_{\gamma, \gamma}(A(\hbar - \varpi)^{\gamma}) \kappa(\varpi, \chi(\varpi), \chi(\varpi - \tau)) d\varpi \\ & + \int_0^{\hbar} (\hbar - \varpi)^{\gamma - 1} E_{\gamma, \gamma}(A(\hbar - \varpi)^{\gamma}) \phi(\varpi, \chi(\varpi), \chi(\varpi - \tau)) dB_{\varpi}. \end{aligned}$$

$\square$

Next, we aim to demonstrate the existence and uniqueness of the solution. In order to get the desired result, we use the Picard iteration technique and the contradiction method.

We define the norm as $\|\chi\|_c = \sup\limits_{\hbar \in [-\tau, V]} \mathbb{E}\|\chi(\hbar)\| < \infty$ on Banach space $C([-\tau, V]; \mathbb{R}^d)$. In addition, we let $P = \max\limits_{0 \leq \hbar \leq V} \|E_{\gamma, \gamma}(A\hbar^{\gamma})\|$.

**Theorem 1.** *Assuming the above hypotheses $(H_1)$ and $(H_2)$ are true, then System (1) has solutions in $C([-\tau, V]; \mathbb{R}^d)$.*

**Proof.** In this proof, we verify the existence of solutions. Define $\chi_0(\hbar) = \chi_0 = \Psi(0)$. Assume $V > 0$ is sufficiently small and satisfies $M = \frac{4(V+4)P^2\mu_1 V^{2\gamma-1}}{2\gamma-1} < \frac{1}{2}$.

Then, we use the Picard iteration technique and write the stochastic process $\{\chi_n(\hbar), n \geq 0\}$ as follows

$$\begin{cases} \chi(\hbar) = & \Psi(\hbar), \hbar \in [-\tau, 0], \\ \chi_{n+1}(\hbar) = & \int_0^{\hbar}(\hbar-\varpi)^{\gamma-1}E_{\gamma,\gamma}(A(\hbar-\varpi)^{\gamma})\kappa(\varpi,\chi_n(\varpi),\chi_n(\varpi-\tau))d\varpi \\ & + \int_0^{\hbar}(\hbar-\varpi)^{\gamma-1}E_{\gamma,\gamma}(A(\hbar-\varpi)^{\gamma})\sigma(\varpi,\chi_n(\varpi),\chi_n(\varpi-\tau))dB_{\varpi}, \\ & \hbar \in [0, V]. \end{cases} \tag{3}$$

**Step 1.** Prove that the sequence $\{\chi_n(\hbar)\}$ is bounded. By using the Jensen inequality, we can obtain

$$\begin{aligned} &\sup_{0 \leq \hbar \leq V} \mathbb{E}\|\chi_{n+1}(\hbar)\|^2 \\ &\leq 2 \sup_{0 \leq \hbar \leq V} \mathbb{E}\left\|\int_0^{\hbar}(\hbar-\varpi)^{\gamma-1}E_{\gamma,\gamma}(A(\hbar-\varpi)^{\gamma})\kappa(\varpi,\chi_n(\varpi),\chi_n(\varpi-\tau))d\varpi\right\|^2 \\ &+ 2 \sup_{0 \leq \hbar \leq V} \mathbb{E}\left\|\int_0^{\hbar}(\hbar-\varpi)^{\gamma-1}E_{\gamma,\gamma}(A(\hbar-\varpi)^{\gamma})\sigma(\varpi,\chi_n(\varpi),\chi_n(\varpi-\tau))dB_{\varpi}\right\|^2 \\ &:= 2(I_1 + I_2). \end{aligned} \tag{4}$$

From the Hölder inequality and assumption $(H_2)$, it is simple to find that

$$\begin{aligned} I_1 &= \sup_{0 \leq \hbar \leq V} \mathbb{E}\left\|\int_0^{\hbar}(\hbar-\varpi)^{\gamma-1}E_{\gamma,\gamma}(A(\hbar-\varpi)^{\gamma})\kappa(\varpi,\chi_n(\varpi),\chi_n(\varpi-\tau))d\varpi\right\|^2 \\ &\leq P^2 \sup_{0 \leq \hbar \leq V} \int_0^{\hbar}(\hbar-\varpi)^{2\gamma-2}d\varpi \cdot \int_0^{\hbar} \mathbb{E}\|\kappa(\varpi,\chi_n(\varpi),\chi_n(\varpi-\tau))\|^2 d\varpi \\ &\leq P^2 \frac{V^{2\gamma-1}}{2\gamma-1} \cdot \sup_{0 \leq \hbar \leq V} \int_0^{\hbar} \mu_2(1 + \mathbb{E}\|\chi_n(\varpi)\|^2 + \mathbb{E}\|\chi_n(\varpi-\tau)\|^2)d\varpi \\ &\leq P^2 \mu_2 \frac{V^{2\gamma-1}}{2\gamma-1} \cdot \left(V + \int_0^{V} \mathbb{E}\sup_{0 \leq r \leq \varpi}\|\chi_n(r)\|^2 d\varpi + \int_0^{V} \mathbb{E}\sup_{0 \leq r \leq \varpi}\|\psi_n(r-\tau)\|^2 d\varpi\right). \end{aligned} \tag{5}$$

By means of hypothesis $(H_2)$ and the Burkholder–Davis–Gundy (B-D-G) inequality, we can derive

$$\begin{aligned} I_2 &= \sup_{0 \leq \hbar \leq V} \mathbb{E}\left\|\int_0^{\hbar}(\hbar-\varpi)^{\gamma-1}E_{\gamma,\gamma}(A(\hbar-\varpi)^{\gamma})\sigma(\varpi,\chi_n(\varpi),\chi_n(\varpi-\tau))dB_{\varpi}\right\|^2 \\ &\leq 4P^2\mathbb{E}\int_0^{V}(V-\varpi)^{2\gamma-2}\|\sigma(\varpi,\chi_n(\varpi),\chi_n(\varpi-\tau))\|^2 d\varpi \\ &\leq 4P^2\int_0^{V}(V-\varpi)^{2\gamma-2}\mu_2(1 + \mathbb{E}\|\chi_n(\varpi)\|^2 + \mathbb{E}\|\chi_n(\varpi-\tau)\|^2)d\varpi \\ &\leq 4P^2\mu_2\left[\frac{V^{2\gamma-1}}{2\gamma-1} + \int_0^{V}\sup_{0 \leq r \leq \varpi}(V-r)^{2\gamma-2}\mathbb{E}\|\chi_n(r)\|^2 d\varpi \right. \\ &\left. + \int_0^{V}\sup_{0 \leq r \leq \varpi}(V-r)^{2\gamma-2}\mathbb{E}\|\chi_n(r-\tau)\|^2 d\varpi\right]. \end{aligned} \tag{6}$$

In general, with Equations (4)–(6), we have

$$\sup_{0\leq\hbar\leq V}\mathbb{E}\|\chi_{n+1}(\hbar)\|^2$$

$$\leq 2P^2\mu_2\frac{V^{2\gamma}}{2\gamma-1}+2P^2\mu_2\frac{V^{2\gamma-1}}{2\gamma-1}\int_0^V\mathbb{E}\sup_{0\leq r\leq\varpi}\|\chi_n(r)\|^2d\varpi$$

$$+2P^2\mu_2\frac{V^{2\gamma-1}}{2\gamma-1}\int_0^V\mathbb{E}\sup_{0\leq r\leq\varpi}\|\chi_n(r-\tau)\|^2d\varpi$$

$$+8P^2\mu_2\frac{V^{2\gamma-1}}{2\gamma-1}+8P^2\mu_2\int_0^V\sup_{0\leq r\leq\varpi}(V-r)^{2\gamma-2}\mathbb{E}\|\chi_n(r)\|^2d\varpi$$

$$+8P^2\mu_2\int_0^V\sup_{0\leq r\leq\varpi}(V-r)^{2\gamma-2}\mathbb{E}\|\chi_n(r-\tau)\|^2d\varpi$$

$$\leq 2P^2\mu_2\frac{V^{2\gamma-1}}{2\gamma-1}(V+4)+4P^2\mu_2\frac{V^{2\gamma}}{2\gamma-1}\|\chi_n\|_c^2+16P^2\mu_2\frac{V^{2\gamma-1}}{2\gamma-1}\|\chi_n\|_c^2.$$

Subsequently, we can draw a conclusion that there is a constant $C$ that satisfies

$$\sup_{0\leq\hbar\leq V}\mathbb{E}\|\chi_{n+1}(\hbar)\|^2\leq C.$$

**Step 2.** Prove the sequence $\{\chi_n(\hbar)\}$ is a Cauchy sequence. From Equation (3), we get the following formula

$$\chi_{n+1}(\hbar)-\chi_n(\hbar)$$

$$=\int_0^{\hbar}(\hbar-\varpi)^{\gamma-1}E_{\gamma,\gamma}(A(\hbar-\varpi)^{\gamma})[\kappa(\varpi,\chi_n(\varpi),\chi_n(\varpi-\tau))-\kappa(\varpi,\chi_{n-1}(\varpi),\chi_{n-1}(\varpi-\tau))]d\varpi$$

$$+\int_0^{\hbar}(\hbar-\varpi)^{\gamma-1}E_{\gamma,\gamma}(A(\hbar-\varpi)^{\gamma})\cdot[\sigma(\varpi,\chi_n(\varpi),\chi_n(\varpi-\tau))-\sigma(\varpi,\chi_{n-1}(\varpi),\chi_{n-1}(\varpi-\tau))]dB_{\varpi}.$$

In particular, for $n=0$, from $(H_2)$ and the B-D-G inequality, Jensen inequality, and Hölder inequality, we have

$$\mathbb{E}\sup_{0\leq\widetilde{\hbar}\leq\hbar}\left\|\chi_1(\widetilde{\hbar})-\chi_0(\widetilde{\hbar})\right\|^2$$

$$=\mathbb{E}\sup_{0\leq\widetilde{\hbar}\leq\hbar}\|\int_0^{\widetilde{\hbar}}(\widetilde{\hbar}-\varpi)^{\gamma-1}E_{\gamma,\gamma}(A(\widetilde{\hbar}-\varpi)^{\gamma})\kappa(\varpi,\chi_0(\varpi),\chi_0(\varpi-\tau))d\varpi$$

$$+\int_0^{\widetilde{\hbar}}(\widetilde{\hbar}-\varpi)^{\gamma-1}E_{\gamma,\gamma}(A(\widetilde{\hbar}-\varpi)^{\gamma})\sigma(\varpi,\chi_0(\varpi),\chi_0(\varpi-\tau))dB_{\varpi}-\Psi(0)\|^2$$

$$\leq 3\mathbb{E}\sup_{0\leq\widetilde{\hbar}\leq\hbar}\|\Psi(0)\|^2$$

$$+3\mathbb{E}\sup_{0\leq\widetilde{\hbar}\leq\hbar}\left\|\int_0^{\widetilde{\hbar}}(\widetilde{\hbar}-\varpi)^{\gamma-1}E_{\gamma,\gamma}(A(\widetilde{\hbar}-\varpi)^{\gamma})\cdot\kappa(\varpi,\chi_0(\varpi),\chi_0(\varpi-\tau))d\varpi\right\|^2$$

$$+3\mathbb{E}\sup_{0\leq\widetilde{\hbar}\leq\hbar}\left\|\int_0^{\widetilde{\hbar}}(\widetilde{\hbar}-\varpi)^{\gamma-1}E_{\gamma,\gamma}(A(\widetilde{\hbar}-\varpi)^{\gamma})\cdot\sigma(\varpi,\chi_0(\varpi),\chi_0(\varpi-\tau))dB_{\varpi}\right\|^2$$

$$\leq 3\Psi^2(0)+3P^2V\cdot\mathbb{E}\int_0^{\hbar}(\hbar-\varpi)^{2\gamma-2}\|\kappa(\varpi,\chi_0(\varpi),\chi_0(\varpi-\tau))\|^2d\varpi$$

$$+12P^2\cdot\mathbb{E}\int_0^{\hbar}(\hbar-\varpi)^{2\gamma-2}\|\sigma(\varpi,\chi_0(\varpi),\chi_0(\varpi-\tau))\|^2d\varpi$$

$$\leq 3\Psi^2(0)+3P^2V\mu_2\cdot\int_0^{\hbar}(\hbar-\varpi)^{2\gamma-2}(1+\mathbb{E}\|\chi_0(\varpi)\|^2+\mathbb{E}\|\chi_0(\varpi-\tau)\|^2)d\varpi$$

$$+12P^2\mu_2\cdot\int_0^{\hbar}(\hbar-\varpi)^{2\gamma-2}(1+\mathbb{E}\|\chi_0(\varpi)\|^2+\mathbb{E}\|\chi_0(\varpi-\tau)\|^2)d\varpi$$

$$\leq 3\Psi^2(0)+3P^2V\mu_2\cdot\int_0^{\hbar}(\hbar-\varpi)^{2\gamma-2}(1+2\mathbb{E}\|\chi_0\|_c^2)d\varpi$$

$$+12P^2\mu_2\cdot\int_0^{\hbar}(\hbar-\varpi)^{2\gamma-2}(1+2\mathbb{E}\|\chi_0\|_c^2)d\varpi$$

$$\leq 3\Psi^2(0)+\frac{3P^2\mu_2(V+4)}{2\gamma-1}V^{2\gamma-1}(1+2\mathbb{E}\|\chi_0\|_c^2).$$

It can be found from $(H_1)$ and the B-D-G inequality, Jensen inequality, and Hölder inequality that

$$
\mathbb{E} \sup_{0 \le \widetilde{\hbar} \le \hbar} \left\| \chi_{n+1}(\widetilde{\hbar}) - \chi_n(\widetilde{\hbar}) \right\|^2
$$

$$
\le 2\mathbb{E} \sup_{0 \le \widetilde{\hbar} \le \hbar} \left\| \int_0^{\widetilde{\hbar}} (\widetilde{\hbar} - \varpi)^{\gamma - 1} E_{\gamma,\gamma}(A(\widetilde{\hbar} - \varpi)^\gamma) \right.
$$

$$
\cdot \left. [\kappa(\varpi, \chi_n(\varpi), \chi_n(\varpi - \tau)) - \kappa(\varpi, \chi_{n-1}(\varpi), \chi_{n-1}(\varpi - \tau))] d\varpi \right\|^2
$$

$$
+ 2\mathbb{E} \sup_{0 \le \widetilde{\hbar} \le \hbar} \left\| \int_0^{\widetilde{\hbar}} (\widetilde{\hbar} - \varpi)^{\gamma - 1} E_{\gamma,\gamma}(A(\widetilde{\hbar} - \varpi)^\gamma) \right.
$$

$$
\cdot \left. [\sigma(\varpi, \chi_n(\varpi), \chi_n(\varpi - \tau)) - \sigma(\varpi, \chi_{n-1}(\varpi), \chi_{n-1}(\varpi - \tau))] dB_\varpi \right\|^2
$$

$$
\le 2P^2 V \int_0^{\hbar} (\hbar - \varpi)^{2\gamma - 2}
$$

$$
\cdot \mathbb{E}\|\kappa(\varpi, \chi_n(\varpi), \chi_n(\varpi - \tau)) - \kappa(\varpi, \chi_{n-1}(\varpi), \chi_{n-1}(\varpi - \tau))\|^2 d\varpi \tag{7}
$$

$$
+ 8P^2 \int_0^{\hbar} (\hbar - \varpi)^{2\gamma - 2}
$$

$$
\cdot \mathbb{E}\|\sigma(\varpi, \chi_n(\varpi), \chi_n(\varpi - \tau)) - \sigma(\varpi, \chi_{n-1}(\varpi), \chi_{n-1}(\varpi - \tau))\|^2 d\varpi
$$

$$
\le (2P^2 V + 8P^2)\mu_1 \cdot \int_0^{\hbar} (\hbar - \varpi)^{2\gamma - 2}
$$

$$
\cdot \left( \mathbb{E}\|\chi_n(\varpi) - \chi_{n-1}(\varpi)\|^2 + \mathbb{E}\|\chi_n(\varpi - \tau) - \chi_{n-1}(\varpi - \tau)\|^2 \right) d\varpi
$$

$$
\le 2(2P^2 V + 8P^2)\mu_1 \cdot \int_0^{\hbar} (\hbar - \varpi)^{2\gamma - 2} \mathbb{E}\left( \sup_{0 \le \vartheta \le \varpi} \|\chi_n(\vartheta) - \chi_{n-1}(\vartheta)\|^2 \right) d\varpi
$$

$$
\le \frac{4(V + 4)P^2 \mu_1 V^{2\gamma - 1}}{2\gamma - 1} \mathbb{E}\left( \sup_{0 \le \widetilde{\hbar} \le \hbar} \left\| \chi_n(\widetilde{\hbar}) - \chi_{n-1}(\widetilde{\hbar}) \right\|^2 \right).
$$

Suppose $\mathbb{E} \sup_{0 \le \widetilde{\hbar} \le \hbar} \left\| \chi_n(\widetilde{\hbar}) - \chi_{n-1}(\widetilde{\hbar}) \right\|^2 \le T \cdot M^{n-1}, n = 1, 2, \dots, \hbar$, where

$$
T = \mathbb{E} \sup_{0 \le \widetilde{\hbar} \le \hbar} \left\| \chi_1(\widetilde{\hbar}) - \chi_0(\widetilde{\hbar}) \right\|^2,
$$

$$
M = \frac{4(V + 4)P^2 \mu_1 V^{2\gamma - 1}}{2\gamma - 1},
$$

which are constants and only depend on $\gamma, V, \mu_1$.

It can be obtained from mathematical induction combined with Equation (7) that

$$
\mathbb{E} \sup_{0 \le \widetilde{\hbar} \le \hbar} \left\| \chi_{n+1}(\widetilde{\hbar}) - \chi_n(\widetilde{\hbar}) \right\|^2 \le T \cdot M^n, n = 0, 1, \dots, \hbar,
$$

By means of Chebyshev's inequality, we can obtain

$$
\mathbb{P}\{ \sup_{\widetilde{\hbar} \in [0, \hbar]} \left\| \chi_{n+1}(\widetilde{\hbar}) - \chi_n(\widetilde{\hbar}) \right\|^2 \ge \frac{1}{2^n} \} \le T \cdot (2M)^n.
$$

The sum of both sides of the inequality above is

$$
\sum_{n=0}^{\infty} \mathbb{P}\{ \sup_{\widetilde{\hbar} \in [0, \hbar]} \left\| \chi_{n+1}(\widetilde{\hbar}) - \chi_n(\widetilde{\hbar}) \right\|^2 \ge \frac{1}{2^n} \} \le \sum_{n=0}^{\infty} T \cdot (2M)^n.
$$

We know that $\sum\limits_{n=0}^{\infty} T \cdot (2M)^n < \infty$ by means of the comparison test. Subsequently, the Borel Cantelli lemma can be used to find that $\sup\limits_{\widetilde{\hbar} \in [0,\hbar]} \left\| \chi_{n+1}(\widetilde{\hbar}) - \chi_n(\widetilde{\hbar}) \right\|^2$ converges to 0. This also implies that $\chi_n$ is a Cauchy sequence. Thus, $\chi_n$ converges almost surely and uniform on $[-\tau, V]$ to a limit $\chi(\hbar)$ defined by

$$\lim_{\mathbf{N} \to \infty} \left( \chi_0(\hbar) + \sum_{n=1}^{\mathbf{N}} (\chi_n(\hbar) - \chi_{n-1}(\hbar)) \right) = \lim_{\mathbf{N} \to \infty} \chi_{\mathbf{N}} = \chi(\hbar).$$

Combining the bounds of $\mathbb{E}\|\chi_{n+1}(\hbar)\|^2$ and by using Fatou's lemma, we can obtain

$$\mathbb{E}\|\chi(\hbar)\|^2 \leq 2P^2 \mu_2 \frac{V^{2\gamma-1}}{2\gamma-1}(V+4) + 4P^2 \mu_2 \frac{V^{2\gamma}}{2\gamma-1}\|\chi_n\|_c^2 + 16P^2 \mu_2 \frac{V^{2\gamma-1}}{2\gamma-1}\|\chi_n\|_c^2.$$

Therefore, from Equation (3), we can obtain

$$\chi(\hbar) = \begin{cases} \Psi(\hbar), \hbar \in [-\tau, 0], \\ \int_0^{\hbar} (\hbar-\omega)^{\gamma-1} E_{\gamma,\gamma}(A(\hbar-\omega)^\gamma)\kappa(\omega, \chi(\omega), \chi(\omega-\tau))d\omega \\ + \int_0^{\hbar} (\hbar-\omega)^{\gamma-1} E_{\gamma,\gamma}(A(\hbar-\omega)^\gamma)\sigma(\omega, \chi(\omega), \chi(\omega-\tau))dB_\omega, \\ \omega \in [0, V]. \end{cases} \quad (8)$$

for all $\hbar \in [-\tau, V]$. $\square$

**Theorem 2.** *System* (1) *has a unique solution in* $C([-\tau, V]; \mathbb{R}^d)$ *if hypothesis* $(H_1)$ *holds.*

**Proof.** Here, we use the contradiction method. Suppose System (1) has two different solutions $\chi(\hbar)$ and $\varphi(\hbar)$ and, at the same time, let $\rho(\hbar) = \chi(\hbar) - \varphi(\hbar)$, $\hbar \in [-\tau, V]$. Then, we can obtain the following. For $\hbar \in [-\tau, 0]$, we have

$$\begin{aligned} \rho(\hbar) &= \chi(\hbar) - \varphi(\hbar) \\ &= \Psi(\hbar) - \Psi(\hbar) \\ &= 0; \end{aligned}$$

For $\hbar \in [0, V]$, we have

$$\begin{aligned} \rho(\hbar) &= \int_0^{\hbar} (\hbar-\omega)^{\gamma-1} E_{\gamma,\gamma}(A(\hbar-\omega)^\gamma)[\kappa(\omega, \chi(\omega), \chi(\omega-\tau)) - \kappa(\omega, \varphi(\omega), \varphi(\omega-\tau))]d\omega \\ &\quad + \int_0^{\hbar} (\hbar-\omega)^{\gamma-1} E_{\gamma,\gamma}(A(\hbar-\omega)^\gamma)[\sigma(\omega, \chi(\omega), \chi(\omega-\tau)) - \sigma(\omega, \varphi(\omega), \varphi(\omega-\tau))]dB_\omega. \end{aligned}$$

By applying $(H_1)$ and the Hölder inequality, Jensen inequality, and B-D-G inequality, we can derive

$$\mathbb{E} \sup_{0 \leq \varpi \leq \hbar} \|\rho(\varpi)\|^2$$

$$\leq 2\mathbb{E} \sup_{0 \leq \varpi \leq \hbar} \| \int_0^{\hbar} (\hbar - \varpi)^{\gamma-1} E_{\gamma,\gamma}(A(\hbar - \varpi)^{\gamma})$$

$$\cdot [\kappa(\varpi, \chi(\varpi), \chi(\varpi - \tau)) - \kappa(\varpi, \varphi(\varpi), \varphi(\varpi - \tau))]d\varpi\|^2$$

$$+ 2\mathbb{E} \sup_{0 \leq \varpi \leq \hbar} \| \int_0^{\hbar} (\hbar - \varpi)^{\gamma-1} E_{\gamma,\gamma}(A(\hbar - \varpi)^{\gamma})$$

$$\cdot [\sigma(\varpi, \chi(\varpi), \chi(\varpi - \tau)) - \sigma(\varpi, \varphi(\varpi), \varphi(\varpi - \tau))]dB_{\varpi}\|^2$$

$$\leq 2VP^2 \sup_{0 \leq \varpi \leq \hbar} \int_0^{\hbar} (\hbar - \varpi)^{2\gamma-2} \mathbb{E}\|\kappa(\varpi, \chi(\varpi), \chi(\varpi - \tau)) - \kappa(\varpi, \varphi(\varpi), \varphi(\varpi - \tau))\|^2 d\varpi$$

$$+ 8P^2 \int_0^{\hbar} (\hbar - \varpi)^{2\gamma-2} \mathbb{E}\|\sigma(\varpi, \chi(\varpi), \chi(\varpi - \tau)) - \sigma(\varpi, \varphi(\varpi), \varphi(\varpi - \tau))\|^2 d\varpi$$

$$\leq 2VP^2\mu_1 \int_0^{\hbar} (\hbar - \varpi)^{2\gamma-2} (\mathbb{E} \sup_{0 \leq \omega \leq \varpi} \|\chi(\omega) - \varphi(\omega)\|^2$$

$$+ \mathbb{E} \sup_{0 \leq \omega \leq \varpi} \|\chi(\omega - \tau) - \varphi(\omega - \tau)\|^2)d\varpi$$

$$+ 8P^2\mu_1 \int_0^{\hbar} (\hbar - \varpi)^{2\gamma-2} (\mathbb{E} \sup_{0 \leq \omega \leq \varpi} \|\chi(\omega) - \varphi(\omega)\|^2$$

$$+ \mathbb{E} \sup_{0 \leq \omega \leq \varpi} \|\chi(\omega - \tau) - \varphi(\omega - \tau)\|^2)d\varpi$$

$$= 2VP^2\mu_1 \int_0^{\hbar} (\hbar - \varpi)^{2\gamma-2} (\mathbb{E} \sup_{0 \leq \omega \leq \varpi} \|\rho(\omega)\|^2 + \mathbb{E} \sup_{0 \leq \omega \leq \varpi} \|\rho(\omega - \tau)\|^2)d\varpi$$

$$+ 8P^2\mu_1 \int_0^{\hbar} (\hbar - \varpi)^{2\gamma-2} (\mathbb{E} \sup_{0 \leq \omega \leq \varpi} \|\rho(\omega)\|^2 + \mathbb{E} \sup_{0 \leq \omega \leq \varpi} \|\rho(\omega - \tau)\|^2)d\varpi$$

Assume $\widehat{\rho}(\hbar) = \mathbb{E} \sup_{\varpi \in [-\tau, \hbar]} \|\rho(\varpi)\|$. Therefore, for $\hbar \in [0, V]$, we have $\mathbb{E} \sup_{\varpi \in [-\tau, \hbar]} \|\rho(\varpi)\|^2$
$\leq (\widehat{\rho}(\varpi))^2$ and $\mathbb{E} \sup_{\varpi \in [0, \hbar]} \|\rho(\varpi - \tau)\|^2 \leq (\widehat{\rho}(\varpi))^2$.

Thus,

$$\mathbb{E} \sup_{0 \leq \varpi \leq \hbar} \|\rho(\varpi)\|^2 \leq 2VP^2\mu_1 \int_0^{\hbar} (\hbar - \varpi)^{2\gamma-2} (\mathbb{E} \sup_{0 \leq \omega \leq \varpi} \|\rho(\omega)\|^2 + \mathbb{E} \sup_{0 \leq \omega \leq \varpi} \|\rho(\omega - \tau)\|^2)d\varpi$$

$$+ 8P^2\mu_1 \int_0^{\hbar} (\hbar - \varpi)^{2\gamma-2} (\mathbb{E} \sup_{0 \leq \omega \leq \varpi} \|\rho(\omega)\|^2 + \mathbb{E} \sup_{0 \leq \omega \leq \varpi} \|\rho(\omega - \tau)\|^2)d\varpi$$

$$\leq 4P^2\mu_1(V + 4) \int_0^{\hbar} (\hbar - \varpi)^{2\gamma-2} (\widehat{\rho}(\varpi))^2 d\varpi.$$

In general, for $\forall \theta \in [0, \hbar]$, we can get

$$\mathbb{E} \sup_{0 \leq \theta \leq \hbar} \|\rho(\theta)\|^2 \leq 4P^2\mu_1(V + 4) \int_0^{\theta} (\theta - \varpi)^{2\gamma-2} (\widehat{\rho}(\varpi))^2 d\varpi$$

$$\leq 4P^2\mu_1(V + 4) \int_0^{\hbar} (\hbar - \varpi)^{2\gamma-2} (\widehat{\rho}(\varpi))^2 d\varpi.$$

Therefore,

$$\|\widehat{\rho}(\hbar)\|^2 = \mathbb{E} \sup_{\varpi \in [-\tau, \hbar]} \|\rho(\varpi)\|^2$$

$$\leq \max\{\mathbb{E} \sup_{\varpi \in [-\tau, 0]} \|\rho(\varpi)\|^2, \mathbb{E} \sup_{\varpi \in [0, \hbar]} \|\rho(\varpi)\|^2\}$$

$$\leq \max\{0, 4P^2\mu_1(V+4) \int_0^{\hbar} (\hbar - \varpi)^{2\gamma - 2} (\widehat{\rho}(\varpi))^2 d\varpi\}$$

$$= 4P^2\mu_1(V+4) \int_0^{\hbar} (\hbar - \varpi)^{2\gamma - 2} (\widehat{\rho}(\varpi))^2 d\varpi.$$

By means of the Grönwall–Bellman inequality, we can get $\|\widehat{\rho}(\hbar)\|^2 \leq 0$. That applies $\rho(\hbar) = 0$. Therefore, System (1) has a unique solution in $C([-\tau, V]; \mathbb{R}^n)$. $\quad\square$

### 4. Finite-Time Stability

**Theorem 3.** *Assume that conditions $(H_1)$ and $(H_2)$ are established and there are positive constants $\varrho$ and $\xi$ satisfying $\varrho < \xi$, $\|\Psi(0)\|^2 \leq \varrho$. We can then deduce that System (1) is finite-time-stable if it satisfies*

$$(4+V)\frac{2P^2\mu_2 V^{2\gamma - 1}}{2\gamma - 1}$$

$$+ \sum_{n=1}^{\infty} \sum_{i=0}^{n} \binom{n}{i} (\frac{4P^2\mu_2 V^{2\gamma - 1}}{2\gamma - 1})^{n-i} \frac{[16P^2\mu_2\Gamma(2\gamma - 1)]^i}{\Gamma(i(2\gamma - 1) + n - i)}$$

$$\cdot \int_0^{\hbar} (\hbar - \varpi)^{\{i(4\gamma - 3) - (i+1-n)\}} (4+V)\frac{2P^2\mu_2 V^{2\gamma - 1}}{2\gamma - 1} d\varpi$$

$$< \xi.$$

**Proof.** From Section 3, we have understood that System (1) has a unique solution, as shown below

$$\chi(\hbar) = \begin{cases} \Psi(\hbar), \hbar \in [-\tau, 0], \\ \int_0^{\hbar} (\hbar - \varpi)^{\gamma - 1} E_{\gamma, \gamma}(A(\hbar - \varpi)^{\gamma})\kappa(\varpi, \chi(\varpi), \chi(\varpi - \tau)) d\varpi \\ + \int_0^{\hbar} (\hbar - \varpi)^{\gamma - 1} E_{\gamma, \gamma}(A(\hbar - \varpi)^{\gamma})\sigma(\varpi, \chi(\varpi), \chi(\varpi - \tau)) dB_{\varpi}, \\ \hbar \in [0, V]. \end{cases} \quad (9)$$

For $\hbar \in [0, V]$, by applying the Jensen inequality, we can obtain

$$\mathbb{E}\left(\sup_{0 \leq \varpi \leq \hbar} \|\chi(\varpi)\|^2\right)$$

$$\leq 2\mathbb{E} \sup_{0 \leq \varpi \leq \hbar} \left\| \int_0^{\hbar} (\hbar - \varpi)^{\gamma - 1} E_{\gamma, \gamma}(A(\hbar - \varpi)^{\gamma})\kappa(\varpi, \chi(\varpi), \chi(\varpi - \tau)) d\varpi \right\|^2$$

$$+ 2\mathbb{E} \sup_{0 \leq \varpi \leq \hbar} \left\| \int_0^{\hbar} (\hbar - \varpi)^{\gamma - 1} E_{\gamma, \gamma}(A(\hbar - \varpi)^{\gamma})\sigma(\varpi, \chi(\varpi), \chi(\varpi - \tau)) dB_{\varpi} \right\|^2$$

$$:= J_1 + J_2.$$

This is given by combining the Hölder inequality and $(H_2)$

$$J_1 = 2\mathbb{E} \sup_{0 \leq \varpi \leq \hbar} \left\| \int_0^\hbar (\hbar - \varpi)^{\gamma-1} E_{\gamma,\gamma}(A(\hbar - \varpi)^\gamma) \kappa(\varpi, \chi(\varpi), \chi(\varpi - \tau)) d\varpi \right\|^2$$

$$\leq 2P^2 \mathbb{E} \sup_{0 \leq \varpi \leq \hbar} \int_0^\hbar (\hbar - \varpi)^{2\gamma-2} d\varpi \cdot \int_0^\hbar \| \kappa(\varpi, \chi(\varpi), \chi(\varpi - \tau)) \|^2 d\varpi \qquad (10)$$

$$\leq \frac{2P^2 \hbar^{2\gamma-1}}{2\gamma-1} \int_0^\hbar \mu_2 (1 + \mathbb{E} \sup_{0 \leq \overline{\varpi} \leq \varpi} \| \psi(\overline{\varpi}) \|^2 + \mathbb{E} \sup_{0 \leq \overline{\varpi} \leq \varpi} \| \psi(\overline{\varpi} - \tau) \|^2) d\varpi.$$

From the B-D-G inequality and by using hypothesis $(H_2)$ again, we can derive

$$J_2 = 2\mathbb{E} \sup_{0 \leq \varpi \leq \hbar} \left\| \int_0^\hbar (\hbar - \varpi)^{\gamma-1} E_{\gamma,\gamma}(A(\hbar - \varpi)^\gamma) \sigma(\varpi, \chi(\varpi), \chi(\varpi - \tau)) dB_\varpi \right\|^2$$

$$\leq 8P^2 \mathbb{E} \int_0^\hbar (\hbar - \varpi)^{2\gamma-2} \| \sigma(\varpi, \chi(\varpi), \chi(\varpi - \tau)) \|^2 d\varpi \qquad (11)$$

$$\leq 8P^2 \int_0^\hbar (\hbar - \varpi)^{2\gamma-2} \mu_2 (1 + \mathbb{E} \sup_{0 \leq \overline{\varpi} \leq \varpi} \| \psi(\overline{\varpi}) \|^2 + \mathbb{E} \sup_{0 \leq \overline{\varpi} \leq \varpi} \| \psi(\overline{\varpi} - \tau) \|^2) d\varpi.$$

Thus, from Equations (10) and (11), we can deduce that

$$\mathbb{E} \left( \sup_{0 \leq \varpi \leq \hbar} \| \chi(\varpi) \|^2 \right)$$

$$\leq 2\mathbb{E} \sup_{0 \leq \varpi \leq \hbar} \left\| \int_0^\hbar (\hbar - \varpi)^{\gamma-1} E_{\gamma,\gamma}(A(\hbar - \varpi)^\gamma) \kappa(\varpi, \chi(\varpi), \chi(\varpi - \tau)) d\varpi \right\|^2$$

$$+ 2\mathbb{E} \sup_{0 \leq \varpi \leq \hbar} \left\| \int_0^\hbar (\hbar - \varpi)^{\gamma-1} E_{\gamma,\gamma}(A(\hbar - \varpi)^\gamma) \sigma(\varpi, \chi(\varpi), \chi(\varpi - \tau)) dB_\varpi \right\|^2$$

$$\leq \frac{2P^2 \hbar^{2\gamma-1}}{2\gamma-1} \int_0^\hbar \mu_2 (1 + \mathbb{E} \sup_{0 \leq \overline{\varpi} \leq \varpi} \| \chi(\overline{\varpi}) \|^2 + \mathbb{E} \sup_{0 \leq \overline{\varpi} \leq \nu} \| \chi(\overline{\varpi} - \tau) \|^2) d\varpi$$

$$+ 8P^2 \int_0^\hbar (\hbar - \varpi)^{2\gamma-2} \mu_2 (1 + \mathbb{E} \sup_{0 \leq \overline{\varpi} \leq \varpi} \| \chi(\overline{\varpi}) \|^2 + \mathbb{E} \sup_{0 \leq \overline{\varpi} \leq \varpi} \| \chi(\overline{\varpi} - \tau) \|^2) d\varpi$$

$$\leq \frac{8P^2 \mu_2 V^{2\gamma-1}}{2\gamma-1}$$

$$+ \frac{2P^2 \mu_2 V^{2\gamma-1}}{2\gamma-1} \int_0^\hbar (1 + \mathbb{E} \| \chi(\varpi) \|^2 + \mathbb{E} \| \chi(\varpi - \tau) \|^2) d\varpi$$

$$+ 8P^2 \mu_2 \int_0^\hbar (\hbar - \varpi)^{2\gamma-2} (\mathbb{E} \| \chi(\varpi) \|^2 + \mathbb{E} \| \chi(\varpi - \tau) \|^2) d\varpi.$$

Letting $\Theta(\hbar) = \sup_{-\tau \leq \alpha \leq \hbar} \| \chi(\alpha) \|^2$, we can then derive

$$\Theta(\nu) = \sup_{-\tau \leq \alpha \leq \varpi} \| \chi(\alpha) \|^2 \geq \| \chi(\varpi) \|^2,$$

and

$$\| \chi(\varpi - \tau) \|^2 \leq \sup_{-\tau \leq \alpha \leq \varpi - \tau} \| \chi(\alpha) \|^2$$

$$\leq \sup_{-\tau \leq \alpha \leq \varpi} \| \chi(\alpha) \|^2$$

$$= \Theta(\varpi).$$

Thus,

$$\| \chi(\varpi) \|^2 \leq \Theta(\varpi),$$

and

$$\|\chi(\varpi - \tau)\|^2 \leq \Theta(\varpi).$$

Then, we have

$$\mathbb{E}\left(\sup_{0 \leq \varpi \leq \hbar} \|\chi(\varpi)\|^2\right)$$

$$\leq \frac{8P^2\mu_2 V^{2\gamma - 1}}{2\gamma - 1}$$

$$+ \frac{2P^2\mu_2 V^{2\gamma - 1}}{2\gamma - 1} \int_0^\hbar (1 + \mathbb{E}\|\chi(\varpi)\|^2 + \mathbb{E}\|\chi(\varpi - \tau)\|^2) d\varpi$$

$$+ 8P^2\mu_2 \int_0^\hbar (\hbar - \varpi)^{2\gamma - 2} (\mathbb{E}\|\chi(\varpi)\|^2 + \mathbb{E}\|\chi(\varpi - \tau)\|^2) d\varpi$$

$$\leq \frac{8P^2\mu_2 V^{2\gamma - 1}}{2\gamma - 1}$$

$$+ \frac{2P^2\mu_2 V^{2\gamma - 1}}{2\gamma - 1} \int_0^\hbar (1 + \mathbb{E}(\Theta(\varpi)) + \mathbb{E}(\Theta(\varpi))) d\varpi$$

$$+ 8P^2\mu_2 \int_0^\hbar (\hbar - \varpi)^{2\gamma - 2} (\mathbb{E}(\Theta(\varpi)) + \mathbb{E}(\Theta(\varpi))) d\varpi$$

$$\leq (4 + V) \frac{2P^2\mu_2 V^{2\gamma - 1}}{2\gamma - 1}$$

$$+ \frac{2P^2\mu_2 V^{2\gamma - 1}}{2\gamma - 1} \int_0^\hbar 2\mathbb{E}(\Theta(\varpi)) d\varpi$$

$$+ 8P^2\mu_2 \int_0^\hbar (\hbar - \varpi)^{2\gamma - 2} \cdot 2\mathbb{E}(\Theta(\varpi)) d\varpi.$$

For all $\alpha \in [0, \hbar]$, we have

$$\mathbb{E}\left(\sup_{0 \leq \varpi \leq \alpha} \|\chi(\varpi)\|^2\right)$$

$$\leq (4 + V) \frac{2P^2\mu_2 V^{2\gamma - 1}}{2\gamma - 1} + \frac{2P^2\mu_2 V^{2\gamma - 1}}{2\gamma - 1} \int_0^\alpha 2\mathbb{E}(\Theta(\varpi)) d\varpi$$

$$+ 8P^2\mu_2 \int_0^\alpha (\hbar - \varpi)^{2\gamma - 2} \cdot 2\mathbb{E}(\Theta(\varpi)) d\varpi$$

$$\leq (4 + V) \frac{2P^2\mu_2 V^{2\gamma - 1}}{2\gamma - 1} + \frac{2P^2\mu_2 V^{2\gamma - 1}}{2\gamma - 1} \int_0^\hbar 2\mathbb{E}(\Theta(\varpi)) d\varpi$$

$$+ 8P^2\mu_2 \int_0^\hbar (\hbar - \varpi)^{2\gamma - 2} \cdot 2\mathbb{E}(\Theta(\varpi)) d\varpi.$$

Therefore,

$$\mathbb{E}(\Theta(\hbar))$$

$$= \mathbb{E} \sup_{-\tau \leq \alpha \leq \hbar} \|\chi(\alpha)\|^2$$

$$\leq \max\{\mathbb{E} \sup_{-\tau \leq \alpha \leq 0} \|\chi(\alpha)\|^2, \mathbb{E} \sup_{0 \leq \alpha \leq \hbar} \|\chi(\alpha)\|^2\}$$

$$\leq \max\{\varrho, (4 + V) \frac{2P^2\mu_2 V^{2\gamma - 1}}{2\gamma - 1} + \frac{4P^2\mu_2 V^{2\gamma - 1}}{2\gamma - 1} \int_0^\hbar \mathbb{E}(\Theta(\varpi)) d\varpi$$

$$+ 16P^2\mu_2 \int_0^\hbar (\hbar - \varpi)^{2\gamma - 2} \cdot \mathbb{E}(\Theta(\varpi)) d\varpi\}.$$

$\square$

By applying Lemma 6, and according to the condition given by Theorem 3, we can obtain

$$\mathbb{E}(\Theta(\hbar))$$
$$\leq (4+V)\frac{2P^2\mu_2 V^{2\gamma-1}}{2\gamma-1}$$
$$+\sum_{n=1}^{\infty}\sum_{i=0}^{n}\binom{n}{i}(\frac{4P^2\mu_2 V^{2\gamma-1}}{2\gamma-1})^{n-i}\frac{[16P^2\mu_2\Gamma(2\gamma-1)]^i}{\Gamma(i(2\gamma-1)+n-i)}$$
$$\cdot\int_0^{\hbar}(\hbar-\varpi)^{\{i(4\gamma-3)-(i+1-n)\}}(4+V)\frac{2P^2\mu_2 V^{2\gamma-1}}{2\gamma-1}d\varpi$$
$$<\xi.$$

Therefore,

$$\mathbb{E}\left(\sup_{0\leq\varpi\leq\alpha}\|\chi(\varpi)\|^2\right)\leq\mathbb{E}\left(\sup_{-\tau\leq\alpha\leq\hbar}\|\chi(\alpha)\|^2\right)=\mathbb{E}(\Theta(\hbar))<\xi.$$

We can thus know that System (1) is finite-time-stable on $[-\tau, V]$ from Definition 5.

## 5. Example

Consider the equations given below

$$\begin{cases} D_{0^+}^{0.8,0.5}\chi(\hbar) = & A\chi(\hbar)+0.01e^{-\hbar}\chi(\hbar)+0.01\chi(\hbar-\tau) \\ & +(0.01\cos(\hbar)\chi(\hbar)+0.01\sin(\hbar)\chi(\hbar-\tau))\frac{dB_\hbar}{d\hbar}, \hbar\in[0,10], \\ \chi(\hbar)=\Psi(\hbar), & -\tau\leq\hbar\leq 0, \\ I_{0^+}^{0.1}\chi(0)=0. \end{cases} \quad (12)$$

By simple calculation, we can get $\mu_1 = 0.0002$ and $\mu_2 = 0.0002$.

Let $\varrho = 0.1$, $\xi = 0.3$, and $A = 0.1\mathfrak{I}$ ($\mathfrak{I}$ is the one-dimensional identity matrix). It can be obtained by using mathematical software that

$$P = \max_{0\leq\hbar\leq V}\|E_{\gamma,\gamma}(A\hbar^\gamma)\| = 2.0224.$$

We can find that the assumptions of Theorems 1 and 3 are satisfied through simple verification. Thus, System (12) has a unique solution, and we can conclude that the norm of the solution does not reach beyond $\xi = 0.3$ above $[0, 10]$. From Definition 5, the conclusion that System (12) is finite-time-stable on $[0, 10]$ can be drawn.

## 6. Conclusions

In this manuscript, we focused on the existence, uniqueness, and finite-time stability of a kind of Hilfer-type fractional stochastic system with delay. Firstly, the existence of solutions was derived using the Picard iteration technique. Secondly, the uniqueness result was obtained using the the contradiction method. Subsequently, the finite-time stability was obtained by combining the Hölder inequality, B-D-G inequality, Jensen inequality, and generalized Grönwall–Bellman inequality. Finally, an example was provided to account for the validity of the theoretical results.

In the future, the focus of our research will be fuzzy differential equations. How to deal with random terms in fuzzy differential equations has been an inconvenient point in our following work and this will make our research more interesting.

**Author Contributions:** All authors of this paper participated in the exploration and in obtaining the main results, and the contributions of all authors were equal. All authors have read and agreed to the published version of the manuscript.

**Funding:** This research was funded by the Doctoral Research Start-up Fund Project of Nanyang Institute of Technology, the Interdisciplinary Sciences Project of Nanyang Institute of Technology, the Natural Science Special Research Fund Project of Guizhou University, China (202002), and the APC was funded by the Doctoral Research Start-up Fund Project of Nanyang Institute of Technology, the Interdisciplinary Sciences Project of Nanyang Institute of Technology, the Natural Science Special Research Fund Project of Guizhou University, China (202002).

**Data Availability Statement:** No data were used for the research described in the article.

**Acknowledgments:** All the authors of this manuscript would like to express their sincere thanks to the editors and reviewers for their valuable comments, which will also improve the quality of our articles.

**Conflicts of Interest:** The authors declare no conflict of interest.

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
