# Peer review of "A Result Regarding Finite-Time Stability for Hilfer Fractional Stochastic Differential Equations with Delay"

_fractalfract, doi:10.3390/fractalfract7080622_

Round 1

Reviewer 1 Report

Comments on the manuscript #: mathematics-2423181 Titled:

A result on finite-time stability for Hilfer fractional stochastic differential equations with delay

by

Man Li , Yujun Niu , Jing Zou 

The manuscript you are referring to discusses a specific type of stochastic differential equation called a Hilfer fractional stochastic differential equation with delay. The authors employ several mathematical techniques to analyze the behavior of solutions to this equation.

Firstly, they use Laplace transforms to obtain solutions to the equation. Laplace transforms are a mathematical tool used to transform a function of time into a function of a complex variable called the Laplace variable. This technique can be used to obtain analytical solutions to differential equations. Next, the authors use the Picard iteration technique to demonstrate the existence of solutions to the equation. The Picard iteration technique is an iterative method used to obtain solutions

to differential equations by approximating the solution with a sequence of functions. Finally, the authors use the generalized Gronwall-Bellman inequality to obtain finite-time stability of the solutions to the equation. The generalized Gronwall-Bellman inequality is a mathematical inequality used to determine the stability of a system over a finite time interval.

To verify their theoretical results, the authors provide an example illustrating the behavior of solutions to the Hilfer fractional stochastic differential equation with delay. Overall, the manuscript provides a thorough analysis of the behavior of solutions to this specific type of stochastic differential equation, using a range of mathematical techniques.

I see that the results of this manuscript are interesting and deserve publication in Fractal Fract  after taking the following comments into account:-

1.    In page 1 line 5, replace “the finite-time stability” with “ finite-time stability”.

2.    In page 1 line 14, replace “For related contents” with “ For related content”.

3.    In page 1 line 18 and 19 , replace “contents can refer to” with “ that can be referred to in the”.

4.    In page 1 line 20 , replace “caputo” with “ Caputo”.

5.    In page 1 line 22 , replace “Random disturbances is” with “ Random disturbances are”.

6.    In page 1 line 26 , replace “attentions” with “ attention”.

7.    In page 1 line 29 , replace “delay” with “ delays”.

8.    In page 1 line 30 , replace “current states, and its past states” with “ current state, and its past state”.

9.    In page 2 line 57 , replace “have been studies on the finite-time” with “ there were studies on finite-time stability”.

10.In page 2 line 58 , replace “linear systems and discussed the short-time stability of them” with “ linear system and discussed its short-time stability”.

 Finally studying the topic of finite-time stability for Hilfer fractional stochastic differential equations with delay may have limited practical applications in real-world systems, which can be treated by giving more detail in the example section 5   in the tool of solution (give detail), name of the software, and attach code of solution as an appendix.

1.    In page 1 line 5, replace “the finite-time stability” with “ finite-time stability”.

2.    In page 1 line 14, replace “For related contents” with “ For related content”.

3.    In page 1 line 18 and 19 , replace “contents can refer to” with “ that can be referred to in the”.

4.    In page 1 line 20 , replace “caputo” with “ Caputo”.

5.    In page 1 line 22 , replace “Random disturbances is” with “ Random disturbances are”.

6.    In page 1 line 26 , replace “attentions” with “ attention”.

7.    In page 1 line 29 , replace “delay” with “ delays”.

8.    In page 1 line 30 , replace “current states, and its past states” with “ current state, and its past state”.

9.    In page 2 line 57 , replace “have been studies on the finite-time” with “ there were studies on finite-time stability”.

10.In page 2 line 58 , replace “linear systems and discussed the short-time stability of them” with “ linear system and discussed its short-time stability”.

Reviewer 2 Report

In the attached file I indicated some typos and hints to improve your work.

In definition of system (1) it is important to indicate the size of the matrix A!

In Theorem 3.3. you intend to use "the contradiction method", but where is the "desired" contradiction?

An example of  syntax issue is

We derive the equivalent form of system (1) under consideration through the Laplace and its inverse transformation in this section, which is expressed by the Mittag-Leffler function.

This is the first sentence of Section 3.

Reviewer 3 Report

In this paper, finite-time stability for Hilfer fractional stochastic differential equations with delaywas introduced. Numerical examples are provided to demonstrate the accuracy and efficiency of the method.

It seems to me that the presented results are new and interesting. The paper is well organized and well written. However, I have the following suggestions and corrections to improve the current version of this paper.

1)      The authors should include comparisons to other methods.

2)      Grammatical error and some typos exist that should be checked and corrected throughout the paper.

3)      Insert references for all definitions, Lemmas, corollaries and theories.

4)      The references list should be modified.

In this paper, finite-time stability for Hilfer fractional stochastic differential equations with delaywas introduced. Numerical examples are provided to demonstrate the accuracy and efficiency of the method.

It seems to me that the presented results are new and interesting. The paper is well organized and well written. However, I have the following suggestions and corrections to improve the current version of this paper.

1)      The authors should include comparisons to other methods.

2)      Grammatical error and some typos exist that should be checked and corrected throughout the paper.

3)      Insert references for all definitions, Lemmas, corollaries and theories.

4)      The references list should be modified.

Round 2

Reviewer 2 Report

I just wanted to take a moment to express my appreciation for this research paper. Thank you for sharing your knowledge with the world and for making a difference in the scientific community. Your contributions are greatly valued and appreciated.

Reviewer 3 Report

The paper is accepted in the current version